# Noise Was Obviously Reduced by Both Leaf Texture and Surface Roughness in Leaf Scale

**DOI:** 10.3390/plants14091363

**Published:** 2025-04-30

**Authors:** Guangpeng Sun, Bingqian Ma, Xianwen Li, Juyang Liao, Liuduan Wei, Xuan Guo, Chengyang Xu, Xiangqi Kong, Guixiang Jin, Yingshan Jin

**Affiliations:** 1Research Center for Urban Forestry of Beijing Forestry University, Key Laboratory for Silviculture and Forest Conservation of the Ministry of Education, Beijing 100083, China; 2Changsha–Zhuzhou–Xiangtan Long-Term Observation Station of Urban Agglomeration Ecosystem, Changsha 410116, China; 3Southwest Engineering Research Center for Landscape Architecture, Southwestern Forestry University, Kunming 650224, China; 4Jilin Provincial Academy of Forestry Sciences, Changchun 130033, China; 5Beijing Institute of Landscape Architecture, Beijing 100102, China

**Keywords:** noise reduction, leaf texture, leaf surface roughness, leaf tissue density, leaf dry mass per area

## Abstract

(1) Woody plant species selection for urban forests is one of the key factors in reducing traffic noise in urban areas, and the ability of sound wave attenuation by leaves is one of the foundations for species selection. However, less references regarding the relationships between leaf morphological traits and noise reduction have been reported, especially the relationships between leaf texture (LT), leaf surface roughness (LSR), and noise reduction. (2) Eighteen arbors and shrubs were selected based on leaf texture and surface roughness characteristics, and noise reduction was measured using white noise sources in a self-designed device in a quiet laboratory at night. Then, the changes in noise reduction with LT and LSR were analyzed. (3) The noise reduction was significantly affected by LT, LSR, and their interaction (*p* < 0.05). The coriaceous leaf was usually more efficient in noise reduction than the chartaceous leaf, and LSR had an auxiliary effect on noise reduction. The effects of noise reduction were mainly influenced by leaf texture through physical blocking and by leaf surface roughness through interference. (4) The findings demonstrate that leaf texture and leaf surface roughness are the suitable predictors for selecting highly efficient woody plants for establishing and improving noise-reduction-oriented forests.

## 1. Introduction

Noise pollution, which is threatening the health and life quality of human beings [1], has become one of the three ecological priorities in environmental contamination control [2]. According to the World Health Organization (WHO), at least one in five European people have been exposed to noise that can directly cause harmfulness to health [3]. And serious health problems, such as hearing loss [4,5], anxiety and insomnia [6,7], and decrease in women’s fertility [8,9] are caused by noise when people have exposure to a high-intensity noise environment for a long time. Furthermore, traffic accident risk is evidently enhanced by traffic noise via increasing drivers’ mental stress and disturbing drivers’ attention [10]. Noise can also increase the risk of biodiversity conservation within and around urban regions. A great deal of research has proved that noise can disturb birds nesting [11], affect bird physiological and propagation performances [12], impact bird mating and foraging behavior [13], and consequently decrease biodiversity. Thus, reducing urban noise through vegetation development has been one of the global hotspots in recent years [14]. Therefore, mitigating urban noise and providing a quieter environment for the public through urban forest development has become one of the important public concerns in improving human well-being.

Numerical studies on the effects of forests on noise reduction have been published [15], while the inter-tree species differences in noise reduction ability [16,17], sound wave propagation in forests [18], and effects of structures of forest belts on noise attenuation [19,20,21] have been mainly focused on. It has been well known that noise attenuation by forests is mainly affected by height, width, and stock density of tree belts [20,22,23], tree species composition [17], and spatial configuration of trees in forests [22,24]. The horizontal configuration with higher stock density was more effective in noise reduction [20], and forests with multi vertical strata composed of multiple arbors and shrubs were considered as more efficient to attenuate noise compared to mono-cultural forests [25]. This is because acoustic wave propagation usually is disturbed via physical interception, scattering [26], reflection [27] by trunks, branches, and leaves, and absorption and vibration by leaves [28]. Trees with large diameters could efficiently attenuate noise [29]. However, the function of noise reduction caused by crown structure was found to be very weak, and only 0.250~1.76 dB/m of noise was reduced by tree crowns [30].

From the perspective of tree species selection, leaf traits, crown architecture, and morphological traits are crucial factors in noise reduction. In general, broad-leaved trees with wider and thicker leaves, dense foliage [31,32], lower height of crown base [30], and higher branch and/or leaf density per volume of the crown [19,33] are propitious to acoustic attenuation. And noise reduction is significantly affected by leaf area [34] or leaf area index [19], crown size and shape [30], bark characteristics [35], timber density [36] of tree species, and also vegetation cover, for example, NDVI in the local scale of urban forests [37,38].

Leaf plays an important role in noise reduction, especially for broad-leaved tree species. It was reported that leaves in the crown attenuated about 40.9%, 31.0%, and 26.8% of noise in the field when comparing summer noise reduction to winter noise reduction caused by *Platanus orientalis* L., *Robinia pseudoacacia* L., and *Fraxinus rotundifolia* Mill. trees [23], though the results might be affected by complex factors, such as leaf thickness [39], leaf vibration [28,39], and leaf area density (leaf area per crown volume) [33]. Therefore, leaf traits have good potential to predict noise reduction and to be reliable indicators for selecting tree species in the mitigation of noise pollution. It was proved by limited studies that leaf traits, which include leaf size, leaf thickness, petiole characteristics, leaf water contents, and leaf mass, have a significant influence on noise absorption, reflection, and refraction [35,39]. However, as important leaf traits related to sound attenuation, the mechanisms of effects of leaf texture and leaf surface roughness on sound reduction have been unknown. These properties are crucial for sound energy absorption, acoustic wave resistance, and thermal exchange noise absorption [36]. The porous structure on material surfaces notably enhances the shielding effect [40]. For instance, the noise reduction of a perforated metal partition is 1.5 dB higher than that of a conventional sound insulation panel [41]. Thus, we proposed two hypotheses: (1) Physical blocking is the main approach to reduce noise by leaf texture (LT). (2) Disturbance of leaf surface characteristics can enhance noise reduction efficiency.

## 2. Results

### 2.1. Effects of Leaf Texture and Surface Roughness of Leaf on Noise Reduction

Noise reduction was evidently influenced by LT, leaf surface roughness (LSR) and their interaction, and LT had a dominant role in noise reduction compared to LSR (Table 1). The effect of LT on noise reduction was much greater than the effect of LSR because *F* value of LT was 1.5 and 32.0 folds compared to LSR and the interaction between LT and LSR (Table 1). The average relative noise reduction (RNR) caused by coriaceous leaf (CL) was 50.74% higher than that caused by chartaceous leaf (CHL) (Figure 1A) while the differences gradually shrank with an increase in LSR (Figure 1B). However, the noise reduction effect was the most efficient when the LSR was very high, because RNR caused by leaves with rough surfaces on the double epidermis (LSRD) was 44.22% and 40.56% higher than that caused by leaves of leaf surface with smooth surfaces on the double epidermis (LSSD) and leaves with rough surfaces on single epidermis (LSRS), respectively (Figure 1C).

Statistical analysis of the scatter plot demonstrated a significantly positive correlation between leaf tissue density (LTD) and RNR (*p* < 0.05), suggesting that denser leaf tissues may contribute to improved acoustic attenuation performance (Figure 2). The results of Pearson correlation further indicated a highly positive correlation between LTD and RNR (Table 2). Meanwhile, the average LTD values of CL were significantly higher than those of CHL, increasing 9.13%, 23.96%, and 25.04% by leaves with LSSD, LSRS, and LSRD, respectively (Figure 3). Moreover, the correlation coefficient and statistical significance between RNR and CL were notably higher than those between RNR and CHL (Table 2). Therefore, the RNR of CL was significantly higher than that of CHL *(p* < 0.05) (Figure 1A).

LSR exerted a significant effect on RNR (*p* < 0.001; Figure 1C). Furthermore, the RNR caused by CL and CHL showed a synchronous increase with LSR, following a logarithmic trend (Figure 1B). Notably, for CL and CHL, LSR maintained a consistent relationship across different types of leaf surface roughness in the pattern of LSSD < LSRS < LSRD (Figure 4). Correspondingly, the RNR by leaf with LSRD was significantly higher than the leaf with LSRS, which in turn was significantly higher than leaf with LSSD (Figure 1C). Pearson correlation further supported this pattern, showing that the correlation coefficient between LSR and RNR was highest in LSRD, followed by LSRS, and lowest in LSSD (Table 2).

When comparing RNR according to different frequencies of sound wave between CL an CHL, we found that RNR caused by CL were remarkably higher than by CHL when the wave frequencies were below 12 kHz (Figure 5A), and leaves with LSRD were the most efficient in reducing noise waves when the wave frequencies were lower than 12 kHz accordingly (Figure 5B). But there was no significant difference in RNR for leaves with LSSD and LSRS when the wave frequencies were lower than 5 kHz while RNR were reduced more by leaves with LSSD than by leaves with LSRS when the wave frequencies were from 6 kHz to 12 kHz (Figure 5B). The result proved again that CL takes more important role in noise reduction than CHL.

### 2.2. Interactive Effects of Leaf Texture and Leaf Surface Roughness on Noise Reduction

LSR played a significant role in LT-mediated noise attenuation (Figure 6), though the level of interaction between them was relatively limited (Table 1). Because the differences among RNR caused by CL and CHL with LSR were significant (*p* < 0.05), and the maximum difference in RNR led by LRS of CL and CHL were 120.90% and109.65% (Figure 6). However, the differences in RNR caused by CL with LSSD, LRSD and LSRS were higher by 64.9%, 59.6% and 51.2%, respectively, compared to RNR caused by CHL. Thus, the results showed that LT was the dominant factor to noise reduction compared to LSR, and LSR was also important for noise reduction in the leaf.

## 3. Discussion

### 3.1. Effects of Noise Reduction Caused by Leaf Texture Linked with Physical Blocking

Coriaceous leaf is associated with thicker leaves [42] and a higher leaf volume-to-area ratio [43], and the presence of waxy layers and/or cuticle [44]. These morphological traits are further linked to thicker mesophyll tissue and more layers of palisade parenchyma [42]. The noise reduction ability of CL was higher than that of CHL, because the CL had a thicker waxy layer [44], stiffer cell wall [38], higher leaf thickness [39], and significantly higher leaf tissue density (LTD) regardless of the type of LSR in our study (Figure 3). Moreover, RNR clearly increased linearly with LTD. Because the higher value in LTD means that the leaf is thicker and leaf tissue is denser [45], the sound wave propagation distance through a thicker leaf is consequently longer. Denser leaf tissue means that the cellulose density of leaf tissue is usually higher, thus more sound energy is absorbed by denser fibers and porous leaf structure [46]. The conclusion that the stronger effect of noise reduction was closely related to higher green biomass [47] was aligned with the function of LTD in noise reduction. Because the RNR was positively correlated with LTD linearly (Figure 2). LTD was significantly influenced by LT, and LTD of CL was significantly higher than that of CHL regardless of the type of LSR (Figure 3). Consequently, the noise reduction capacity of CL was significantly greater than that of CHL (Figure 1A). Despite being thin, lightweight and having low tissue density, CHL are densely covered with fine pubescence, creating numerous tiny sponge-like spaces. Thus, these characteristics enhance the ability to scatter sound waves, and further increase the energy loss of sound waves, and intensify the refraction effect, thereby promotes the conversion of acoustic energy into thermal energy [28,48,49]. Additionally, the roughness due to large grain bulges further amplifies acoustic wave scattering, indicating that both leaf texture and surface properties influence the noise-reducing effect of leaves. Leaf texture and leaf surface roughness are comprehensively affected by surface characteristics, spatial pattern of vein net, vein diameter, anatomic structure and leaf morphological characteristics.

### 3.2. The Effects of Noise Reduction Caused by Leaf Surface Roughness Linked with Refraction and Interference

Surface roughness is widely recognized as a significant factor influencing the propagation of acoustic waves [48]. A significant positive correlation was observed between LSR and relative noise reduction value (*p* < 0.05) with the leaf noise reduction ability increasing logarithmically as LSR increased (Figure 1B). The structure of leaves with higher LSR exhibited a porous configuration [50], with dense hair coat further enhancing the porosity. This increased porosity likely influenced the noise propagation via thermo-viscous effects [46]. Recent studies have reported that thickness and density of natural fiber are the important factors in sound absorption [46], thus dense bristle on leaf surface should be efficient in noise reduction. Pearson correlation analysis between RNR and surface roughness indicators showed that RNR was extremely correlative to LSR and SRBE (Table 2). Notably, noise reduction attributable to SRBE was significantly greater than that caused by SRDE (Table 2), suggesting that sound waves were intensively interfered by surface contours, protrusions, grooves, and epidermal hairs [51] on the abaxial leaf surface. This indicates that surface roughness could affect sound wave propagation [39]. The biggest surface roughness was observed in CHL with LSRD and followed by CL with LSRD on both the abaxial and adaxial epidermises (Figure 4). However, RNR led by CHL with LSRD was significantly lower than CL with LSRD (Figure 6). Moreover, the Pearson coefficients between RNR and LSR was more significant for CL than for CHL. Notably, only CL with LSRD and CHL with LSRD could explain more variation in RNR, with explanatory powers of 62.4% and 57.5%, respectively. Furthermore, the contribution of LMA to noise reduction was particularly notable in CHL with LSSD and LSRS (Table 2). Therefore, the LT fraction of leaf traits emerged as a critical factor in noise reduction for CHL.

In general, leaves with LSR on the abaxial epidermis exhibited a stronger noise reduction compared to those with LSR localized on the adaxial surface, as evidenced by a higher correlation coefficient between RNR and LSR on the abaxial surface in the complete dataset. (Table 2). This suggested that the oblique incidence of sound waves was induced by metasurface [16] formed by surface roughness of the abaxial epidermis, particularly when noise wave passed leaf from adaxial to the abaxial side.

Actually, chartaceous leaves (CHL) with rough surfaces on double epidermises (LSRD) exhibited the highest surface roughness (Figure 4). Their RNR effects were less relevant to LMA, whereas the RNR effects of CHL with LSRS were closely related to LMA (Table 2). Meanwhile, the result which LTD of all LT and any type of LSR were tightly related to RNR (Table 2), showed that physical blocking of acoustic wave was the main approach to noise reduction. Thus, the noise reduction led by coriaceous leaves was highly dependent on LMA, LTD and independent of LSR, while the noise reduction mediated by chartaceous leaves was dependent on LSR, LMA and LTD, especially for CHL with LSRS (Table 2). However, RNR was strongly dependent on LSR and moderately dependent on LMA for coriaceous leaves when all data of coriaceous leaves were combined. In contrast, RNR was moderately dependent on surface roughness and less related to LMA when all data of chartaceous leaves were integrated. Therefore, these evidences could prove that LT takes the dominant role in noise reduction by leaf, and LSR takes an auxiliary role in noise reduction by leaf. Therefore, leaf traits—specifically leaf texture and leaf surface roughness are the best criteria for selecting woody species with high noise reduction efficiency when establishing and improving noise-reduction-oriented forest. Leaf texture and leaf surface roughness are comprehensively affected by surface characteristics, spatial pattern of the vein net, vein diameter, anatomical structure and overall leaf morphology.

### 3.3. Research Limitations

Leaf texture and leaf surface roughness are comprehensively affected by surface characteristics, spatial patterns of vein network, vein diameter, anatomical structure and leaf morphological characteristics. Noise reduction was extremely affected by interaction between leaf texture and surface roughness, indicating that the results from current research were affected by above mentioned factors of leaf texture and surface roughness. However, we used only a qualitative classification (coriaceous vs. chartaceous). Thus, more quantitative gradients of leaf texture and/or to separate leaf texture into leaf thickness and tissue density, or expressing leaf texture with more relevant leaf traits will be conducted, meanwhile to decouple surface roughness from leaf texture by selecting more woody plant species which include gradients in surface roughness on basis of one leaf texture, and to measure acoustic wave propagation in micro-scale of leaf surface are more important works in the future. Traffic noise frequencies are mainly concentrated between 50 Hz and 5 kHz [52], and the frequencies below 2 kHz were not a focus of our analysis.

## 4. Materials and Methods

### 4.1. Tree Species and Individual Choice

Six tree species were sampled in Changsha Botanical Garden of Hunan Province, China, located on at 113°01′56″ E and 28°06′10″ N, and twelve tree species were sampled in campus of Beijing Forestry University, located at 116°20′85″ E and 40°0′15″ N. All tree species widely used in urban greening were selected according to the characteristics of LT and LSR (Table 3). To avoid variation in leaf traits due to shading, elevated temperatures from impervious surfaces, high stand-density competition, five trees per species growing on the north side of recreational branch roads orienting east–west were selected, and the spacing between trees was 5.0~6.0 m. Sampled trees were located away from large squares, major roads and water bodies. A smooth leaf epidermis was defined as having few epidermal hairs; however, contour, protrusions, and grooves on leaf surface.

### 4.2. Leaves Sampling

To minimize variation in leaf morphological traits induced by differing microenvironments (light and temperature) resulting from leaf positions within the canopy, one-year-old branches on the e southern-facing outer edge of the mid-canopy were harvested with scissors. Because leaf surface characteristics were changed with leaf development [53], fifteen fully expanded leaves, that were healthy and free from pest damage or infection, were collected from the cut branches per tree. Leaves were immediately placed in self-sealing plastic bags, transferred to an insulated container, and transported promptly to the laboratory. All of samples were rinsed with tap water for 1–2 minutes to clean surface dust, blotted dry with absorbent paper, and re-bagged in self-sealing plastic bags, and stored in a refrigerator.

### 4.3. Indoor Noise Reduction Test System

Urban soundscapes are complex [54]; and the majority of urban noise originates from traffic and human conversation [55], whereas natural sounds such as biological sound takes a little proportion [56]. Furthermore, sound pressure measurements remain unaffected when they exceed ambient levels by at least 15 dB [47]. Therefore, a custom-built device was employed to prevent interference from outdoor noise. The device comprised three concentric layers, forming an internal chamber of 600 mm × 600 mm × 1200 mm—ample to accommodate sound sources and sound level meters (Figure 7). The outermost layer consisted of 100 mm-thick sound-insulation panels was made from sound-absorption sponge, effectively attenuating external noise. The middle skeleton layer comprised 20 mm-thick solid wooden boards. The innermost layer combined a 10 mm-thick composite sound insulation board with a 30 mm-thick sound-absorbing sponge to mitigate internal reflections and refractions of the sound waves. A 100 mm-thick sound insulation board with a 30 mm diameter channel served as primary partition, dividing the chamber into two parts: one part was designed to housing the sound source, the sound level meter A (SA) and the sample leaf; the other was designed for a monitoring compartment, where sound level meter B (SB) was positioned at the end opposite sound source (Figure 7).

The sound source center, the SA, the leaf, and the SB were aligned along the horizontal central axis of the channel. The distance between SA and SB was 1000 mm, between sound source and SA, 50 mm; and between SA and the sample leaf, 450 mm. During the test, the leaf was pinned to the side of the central partition facing the sound source to eliminate gaps between the blade and outer edge of the channel, and to minimize leaf vibration.

### 4.4. Noise Reduction Measurement and Calculation

The noise source was white noise generated by Audition CC 2018 (Adobe Systems Inc., San Jose, CA, USA). An AWA6228+sound level meter (Hangzhou Aihua Instrument, Hangzhou, Zhejiang, China), which meets the Class 1 requirements of GB/T3785.1-2010/IEC61672-1:2013 [57], was selected to measure the sound pressure level. The measurement range of AWA6228+ sound level meter was 20–142 dB(A), its frequency response 10 Hz–20 kHz, and its uncertainty 0.4–1 dB. The sound level meter was calibrated using AWA6021A sound calibrator (Hangzhou Aihua Instrument, Hangzhou, Zhejiang, China) before each experiment. The white noise was played through a commercial audio amplifier.

According to the superposition principle for source and ambient noise, the measurement would not be affected when ambient sound pressure levels are at least below the measured sound pressure. We selected 80 dB of sound source intensity [47] which was sufficiently high to prevent interference from ambient noise given that laboratory background levels were approximately 30 dB(A). 

The experiment was conducted at night in a quiet laboratory, away from external traffic noise to minimize ambient interference. Eight reduplicated noise treatments were performed for each leaf. Each treatment lasted 15 s, with a 30s interval between treatments. All readings from sound level meters were automatically recorded inside the closed system to avoid artificial interference during operation.

### 4.5. Determination of Leaf Characteristics

#### 4.5.1. Leaf Segment Sampling for Data Extraction of Surface Characteristics

After the noise reduction experiment, fresh leaves were collected for extraction of surface characteristics data. The central segments of leaf, one was cross midrib and the other was the central part of half blade closed to midrib, were cut off using surgical knife. They measured 3 cm × 3 cm. Then, the leaf segments were scanned to extract surface characteristics data (Figure 8). For each species, ten segments including midrib and ten segments excluding midrib were scanned.

#### 4.5.2. Determination of Leaf Texture

Leaf texture (LT) mainly included three types—coriaceous, chartaceous and membranous leaf [39,58]. LT represents an integrated measure of mechanical properties such as maximum fracture force, toughness, stiffness and displacement [39,59]. Previous studies have classified LT using traits including leaf size, leaf thickness, petiole length, petiole width, petiole thickness and leaf mass. From the perspective of sound wave blocking, variation in LT is influenced by leaf thickness, cuticle thickness, wax layer thickness and leaf tissue density. However, we selected only two easily distinguished LT categories (chartaceous vs. coriaceous) leaf [42] according to the definition of dendrology [39,58] (Table 3).

#### 4.5.3. Data Extraction of Leaf Surface Roughness

Leaf surface roughness is usually caused by surface contour, protrusions, grooves, and epidermal hairs on the lamina [51]. Surface roughness has been difficult to assess by current methods due to limited field depth and restricted observation area across the entire leaf [60], though surface roughness primarily affects acoustic energy attenuation via refraction and interference [51]. Moreover, existing methods were highly dependent on expensive instruments [44], and LSR was characterized at the microscale [61]. Accordingly, we quantified leaf surface roughness following the method of [53]. High-resolution scanner (SZX16) (Olympus Corp., Tokyo, Japan) was then used to capture leaf surface images to analyze LSR (Figure 8). From the perspective of sound wave attenuation, LSR is generated by leaf protrusions, hair, and vein undulations. LSR can be decomposed into surface roughness on adaxial epidermis of lamina (SRDE) and surface roughness on abaxial epidermis of lamina (SRBE). Thus, LSR can be commonly expressed as following:(1)$$\mathrm{LSR}(\%)=\frac{\mathrm{HV}+\mathrm{BA}+\mathrm{VA}}{\mathrm{LA}}\times 100\%,$$(2)$$\mathrm{SRBE}(\%)=\frac{{\mathrm{PA}}_{BE}+{HA}_{BE}+{\mathrm{VA}}_{BE}}{\mathrm{LA}}\times 100\%,$$(3)$$\mathrm{SRDE}(\%)=\frac{{\mathrm{PA}}_{DE}+{HA}_{DE}+{\mathrm{VA}}_{DE}}{\mathrm{LA}}\times 100\%,$$

In the Formula (1), PA was the area of lamina surface protrusion, HA was the area of hair patches, VA was the area of leaf vein, and LA was the leaf area.

In the Formula (2), ${PA}_{BE}$: Protrusion area on abaxial epidermis (BE), ${HA}_{BE}$: Hair area on abaxial epidermis, ${VA}_{BE}$: Vein area on abaxial epidermis, LA: Leaf area

In the Formula (3), ${PA}_{DE}$: Protrusion area on adaxial epidermis (DE), ${HA}_{DE}$: Hair area on adaxial epidermis, ${VA}_{BE}$: Vein area on adaxial epidermis, LA: Leaf area

PA, HA and VA were observed by SZX16 Reflected Fluorescence System (Olympus Corporation, Tokyo, Japan) and calculated through Adobe Photoshop CC 2018 (Adobe Systems Inc., San Jose, CA, USA). The boundaries of hair patches on sparsely pubescent laminae lamina with sparse hair (e.g., *Chimonanthus praecox* (L.) Link) were delineated using the Lasso tool, and patch areas were calculated by counting pixels via Histogram function. For densely hairy laminae, the total area of hairy patches (S_HA_), was estimated from nine randomly sampled circle regions (600-pixel diameter). Leaf hair proportion (LHP) was calculated by counting hair pixels in each region, hair patches area (HA) was then determined using Formula (4). The area of each sampled circle was denoted SC_i_, (i = 1–9), and the hair area in each region as SH_i_, (i = 1–9). HA and LHP for each leaf were then calculated as follows:(4)$$\mathrm{HA}=S_{\mathrm{HA}}\times \mathrm{LHP},$$(5)$$\mathrm{LHP}=\frac{1}{9}\sum_{i=1}^{9}\frac{SH_{i}}{SC_{i}},$$

#### 4.5.4. Leaf Dry Mass per Area

Leaf dry mass per area (LMA, mg·cm^−2^) was calculated as the ratio of dry weight (DW) to leaf area [62]. An increase in LMA typically indicates greater leaf thickness and higher mesophyll tissue density. Therefore, LMA is an important trait characterizing leaf texture [43]. Fresh leaves were scanned using a CanoScan LiDE 300 scanner (Canon Inc., Tokyo, Japan) at 300 dpi to determine leaf area, Leaves were then oven-dried at 105 °C for 5 min, followed by drying at 60 °C for 72 h until constant weight, after which dry mass was measured [63].(6)LMA=DW/LA,

#### 4.5.5. Leaf Tissue Density

Leaf tissue density (LTD, g·cm^−3^) is defined as the ratio of the dry weight to leaf volume (V). LTD usually correlates positively with LMA [64] though the relationship is not strong.(7)LTD=DW/V,

### 4.6. Data Collection and Analysis

#### 4.6.1. Expression of Noise Reduction

Readings from SA and SB were recorded as sound pressure level A (SPL_A_) and sound pressure level B (SPL_B_), the reading from SB was noted as sound pressure level B (SPL_B_), and the value differences between were SA and SB were calculated as the noise reduction value (Figure 7). To isolate leaf-specific effects, a control condition (no leaf) was established: readings from SA and SB was recorded as sound pressure level A_0_ (SPL_A0_) and sound pressure level B_0_ (SPL_B0_) respectively, representing noise reduction due solely to air resistance. Absolute values were applied to ensure that noise-reduction metrics remained positive. Noise reduction ability was calculated as follows:

The relative noise reduction (RNR) value is defined as the difference between noise reduction produced by the leaf-air system in the test group and that produced solely by air in the control group, under same conditions.(8)$$\mathrm{RNR}=\left({\mathrm{SPL}}_{A}\;-\;{\mathrm{SPL}}_{B}\right)\;-\left({\mathrm{SPL}}_{A0}\;-{\mathrm{SPL}}_{B0}\right)$$

#### 4.6.2. Data Analysis

Adobe Photoshop CC 2018 was adopted to extract and quantify leaf surface characteristics. The relative noise reduction ratio and surface roughness of leaves need to be transformed using an arcsine function for subsequent analysis in order to approximate a normal distribution for subsequent significance testing. One-way ANOVA, multiple comparisons, and independent-sample *t*-tests, and Pearson correlation analysis were conducted using the SPSSAU online module (www.spssau.com) of SPSS 25.0 (IBM Corp., Armonk, NY, USA). Line and box charts were produced by GraphPad prism 7.0 (GraphPad Software, San Diego, CA, USA) and SPSS 25.0.

## 5. Conclusions

Noise attenuation performance of woody plant leaves is significantly influenced by leaf texture, leaf surface roughness and the interaction between these two factors. Coriaceous leaf is more efficient than chartaceous leaf in noise reduction, and efficiency of relative noise reduction for both of coriaceous leaf and chartaceous leaf with leaf surface roughness on double epidermis of lamina were the highest, the function of leaf with leaf surface roughness on abaxial epidermis of lamina in noise reduction was stronger when noise wave passed leaf from adaxial epidermis of lamina. In general, the effects of noise reduction are dominantly affected by leaf texture on basis of physical blocking approaches and auxiliary influenced by leaf surface roughness on basis of interference approaches. Our findings identify leaf surface roughness (LSR) and coriaceous texture (LT) as critical micro-scale traits for screening noise-attenuating species. These traits serve as robust morphological indicators for selecting noise-attenuating woody species to optimize noise-reduction-oriented green infrastructure. However, translating trait efficacy to real-world applications requires addressing macro-scale factors—particularly canopy stratification, leaf arrangement synchronicity, and branch porosity—that modulate sound wave scattering. Coupling trait-based selection with multi-layered designs thus provides an ecologically informed strategy to combat urban noise pollution.

## Figures and Tables

**Figure 1 plants-14-01363-f001:**
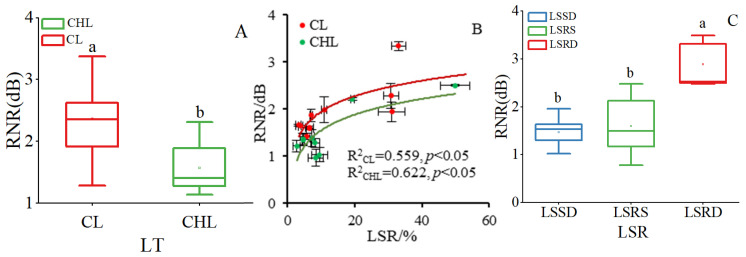
Effect of leaf texture on noise reduction (**A**), relations between leaf surface roughness (LSR) and RNR (**B**), and effect of leaf surface roughness on noise reduction (**C**). RNR, Relative noise reduction value; CL, coriaceous leaf; CHL, chartaceous leaf; LSSD, leaf surface smooth on double epidermis; LSRS, leaf surface roughness on single epidermis; LSRD, leaf surface roughness on double epider-mis. In (**A**), different lowercase letters represent significant effects of LT on RNR (*p* < 0.05). In (**B**), the *x*-axis data of each scatter point represented the mean LTD value of all samples within each tree species, while the *y*-axis data corresponded to the mean RNR value of all samples within each tree species. In (**C**), different lowercase letters represent significant effects of LSR on RNR (*p* < 0.05).

**Figure 2 plants-14-01363-f002:**
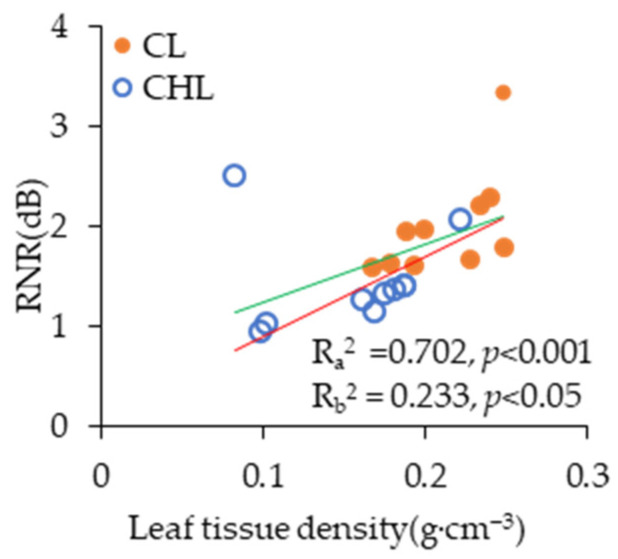
Relations between leaf tissue density (LTD) and RNR. The blue circle at upper location was leaf of *Broussonetia papyrifera* (L.) L’Hér. ex Vent. which was CHL with LSRD, the orange dot was leaf of *Eucommia ulmoides* Oliv. which was CL without LSRD. a: with singular value, red line. b: with singular value, green line. CL, coriaceous leaf; CHL, chartaceous leaf; LSSD, leaf surface smooth on double epidermis; LSRS, leaf surface roughness on single epidermis; LSRD, leaf surface roughness on double epidermis. The *x*-axis data of each scatter point represented the mean LTD value of all samples within each tree species, while the *y*-axis data corresponded to the mean RNR value of all samples within each tree species.

**Figure 3 plants-14-01363-f003:**
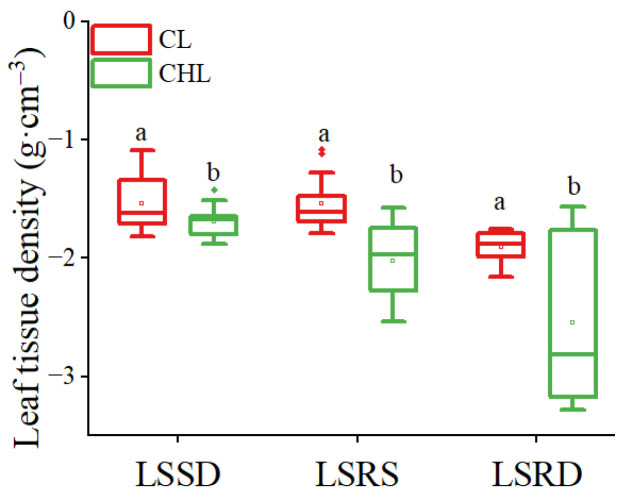
Changes in leaf tissue density with leaf texture and surface characteristics. CL, coriaceous leaf; CHL, chartaceous leaf; LSSD, leaf surface smooth on double epidermis; LSRS, leaf surface roughness on single epidermis; LSRD, leaf surface roughness on double epidermis. The *y*-axis data were the natural logarithm conversion of leaf tissue density. Different lowercase letters represent significant effects of LT on leaf tissue density (*p* < 0.05).

**Figure 4 plants-14-01363-f004:**
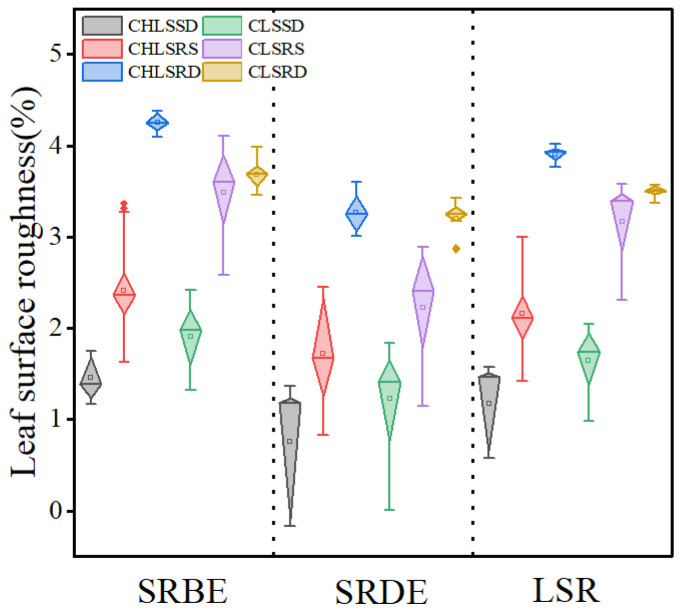
Changes in surface roughness with leaf texture and surface characteristics. CLSSD, coriaceous leaf with smooth on double epidermis; CLSRS, coriaceous leaf with roughness on single epidermis; CLSRD, coriaceous leaf with roughness on double epidermis; CHLSSD, chartaceous leaf with smooth on double epidermis; CHLSRS, chartaceous leaf with roughness on single epidermis; CHLSRD, chartaceous leaf with roughness on double epidermis; SRBE, surface roughness on abaxial epidermis; SRDE, surface roughness on abaxial epidermis; LSR, leaf surface roughness. The *y*-axis data were the natural logarithm conversion of leaf surface roughness.

**Figure 5 plants-14-01363-f005:**
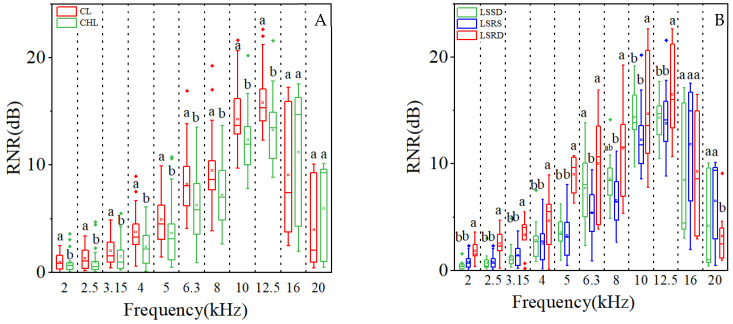
Comparison of relative noise reduction in different spectra caused by leaf textures (**A**) and leaf surface roughness (**B**). CL, coriaceous leaf; CHL, chartaceous leaf; LSSD, leaf surface smooth on double epidermis; LSRS, leaf surface roughness on single epidermis; LSRD, leaf surface roughness on double epidermis. In (**A**), different lowercase letters represent significant effects of LT on RNR (*p* < 0.05). In (**B**), different lowercase letters represent significant difference in RNR between CL and CHL (*p* < 0.05).

**Figure 6 plants-14-01363-f006:**
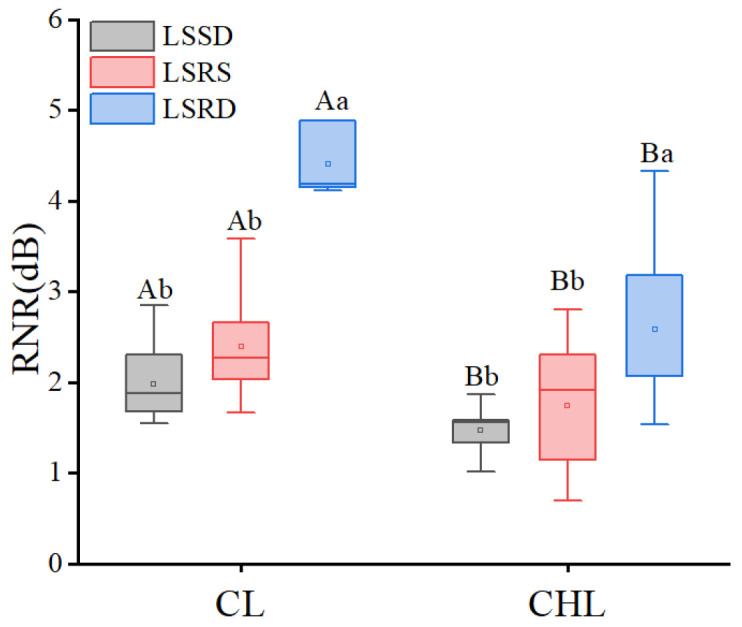
Interactive effects of leaf texture and leaf surface roughness on the noise reduction ability. Different capital letters indicate significant effects of LT on RNR (*p* < 0.05); different lowercase letters indicate significant effects of LSR on RNR (*p* < 0.05). CL, coriaceous leaf; CHL, chartaceous leaf; LSSD, leaf surface smooth on double epidermis; LSRS, leaf surface roughness on single epidermis; LSRD, leaf surface roughness on double epidermis.

**Figure 7 plants-14-01363-f007:**
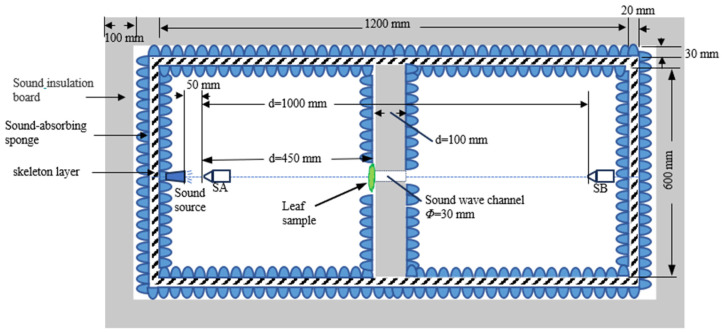
Diagrammatic sketch of device to study single leaf noise reduction. The gray-white regions represent the rigid high-fiber sound insulation board, the blue regions de-note the porous sound-absorbing foam, the black-and-white striped areas correspond to the skeletal template layer, and the green region indicates the test leaf specimen.

**Figure 8 plants-14-01363-f008:**
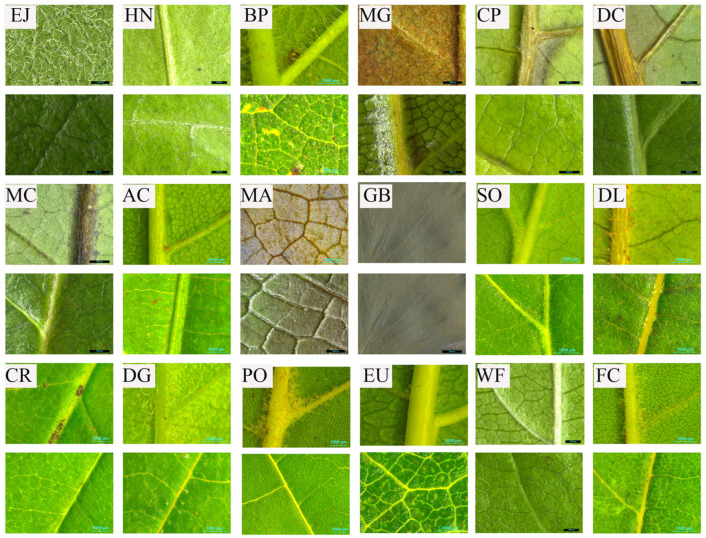
Diagrammatic pictures adaxial and abaxial epidermis of leaves of different species. Notes: EJ, *Eriobotrya japonica* (Thunb.) Lindl.; HN, *Hedera nepalensis var. sinensis* (Tobler) Rehder; BP, *Broussonetia papyrifera* (L.) L’Hér. ex Vent.; MG, *Magnolia grandiflora* L.; CP, *Chimonanthus praecox* (L.) Link; DC, *Daphniphyllum calycinum* Benth.; MC, *Michelia cavaleriei* Finet & Gagnep.; AC, *Aesculus chinensis* Bunge; MA, *Michelia maudiae* Dunn; GB, *Ginkgo biloba* L.; SO, *Syringa oblata* Lindl.; DL, *Diospyros lotus* L.; CR, *Chionanthus retusus* Lindl. & Paxton; DG, *Deutzia grandiflora* Bunge; PO, *Platanus orientalis* L.; EU, *Eucommia ulmoides* Oliv.; WF, *Weigela florida* (Bunge) A. DC.; FC, *Fraxinus chinensis* Roxb.; The scales are 1000 μm. Upper pictures are adaxial epidermis, lower pictures are abaxial epidermis of the same tree species, and the capital letters is abbreviation of tree species.

**Table 1 plants-14-01363-t001:** Variance analysis of leaf texture (LT) and leaf surface roughness (LSR) on relative noise reduction (RNR).

Sources of Variations	RNR	
	F	*p*
LT	118.813	<0.001
LSR	77.776	<0.001
LT × LSR	3.717	0.025

**Table 2 plants-14-01363-t002:** The Pearson correlation between leaf surface roughness (LSR), leaf texture (LT), leaf tissue density (LTD), leaf dry mass per area (LMA) and relative noise reduction (RNR) value. CL, coriaceous leaf; CHL, chartaceous leaf; LSSD, leaf surface smooth on double epidermis; LSRS, leaf surface roughness on single epidermis; LSRD, leaf surface roughness on double epidermis. SRBE, surface roughness on abaxial epidermis of lamina. SRDE, surface roughness on adaxial epidermis of lamina. ns, no significant difference; ^*^, *p* < 0.05; ^**^, *p* < 0.01; ^***^, *p* < 0.001. The data in each cell of the table represented the Pearson correlation coefficient between the structural indicators of and the RNR.

	CL			CHL			LT		All Data
	LSSD	LSRS	LSRD	LSSD	LSRS	LSRD	CL	CHL	
SRDE	0.083 ^ns^	0.389 ^ns^	0.881 ^***^	0.248 ^ns^	0.250 ^ns^	0.977 ^***^	0.748 ^***^	0.668 ^***^	0.645 ^***^
SRBE	0.026 ^ns^	0.496 ^ns^	0.922 ^***^	0.237 ^ns^	0.648 ^***^	0.852 ^***^	0.731 ^***^	0.883 ^***^	0.771 ^***^
LSR	——	0.150 ^ns^	0.790 ^*^	——	0.529 ^*^	0.758 ^***^	0.793 ^***^	0.845 ^***^	0.754 ^***^
LMA	0.479 ^*^	0.564 ^ns^	0.948 ^***^	0.776 ^***^	0.962 ^**^	0.517 ^ns^	0.646 ^***^	0.052 ^ns^	0.500 ^***^
LTD	0.882 ^**^	0.851 ^**^	0.858 ^***^	0.753 ^***^	0.959 ^**^	0.816 ^***^	0.689 ^***^	0.322 ^*^	0.349 ^***^
n	20	15	10	15	20	10	45	45	90

**Table 3 plants-14-01363-t003:** Summary for leaf characteristics of sampled tree and shrub species.

Leaf Texture (LT)		Leaf Surface Roughness (LSR)		Scientific Names of Tree and Shrub Species
Description	Code for Types of LT	Description	Code for Types of LSR	
Coriaceous leaf	CL	smooth on double epidermis	LSSD	*Hedera nepalensis* var. *sinensis* (Tobler) Rehder, *Epipremnum aureum* (Linden & André) Bunting, *Michelia cavaleriei* Finet & Gagnep., *Aesculus chinensis* Bunge, *Michelia maudiae* Dunn, *Syringa oblata* Lindl.
Chartaceous leaf	CHL	smooth on double epidermis	LSSD	*Daphniphyllum calycinum* Benth., *Ginkgo biloba* L.
Coriaceous leaf	CL	roughness on single epidermis	LSRS	*Magnolia grandiflora* L., *Diospyros lotus* L., *Eriobotrya japonica* (Thunb.) Lindl., *Chionanthus retusus* Lindl. & Paxton
Chartaceous leaf	CHL	roughness on single epidermis	LSRS	*Weigela florida* (Bunge) A. DC., *Chimonanthus praecox* (L.) Link, *Deutzia grandiflora* Bunge, *Fraxinus chinensis* Roxb., *Platanus orientalis* L.
Coriaceous leaf	CL	roughness on double epidermis	LSRD	*Eucommia ulmoides* Oliv.
Chartaceous leaf	CHL	roughness on double epidermis	LSRD	*Broussonetia papyrifera* (L.) L’Hér. ex Vent.

## Data Availability

Data will be made available on request.

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
