# Peer review of "Noise Was Obviously Reduced by Both Leaf Texture and Surface Roughness in Leaf Scale"

_plants, 2025, doi:10.3390/plants14091363_

Round 1

Reviewer 1 Report

Comments and Suggestions for Authors

General comments

This ms tested a noise reduction of trees and shrubs in a designed-chamber experiments. It looks good and provides a fundamental information on noise control by plants in a city.

Comment 1

L 301~305, What is a precision and measuring range of device? Please clarify the device description.

Question 2

Based on your data, coriaceous leaf could be an effect at the noise reduction. But, in the field application, macro-scale factors such as leaf arrangement, vacant space of twig and branches could affect the pathway of sound waves. The traits of micro-scale factors of leaf surface roughness and texture could provide a fundamental information to decrease a noise in a city. In the application of your results to a city, what is the priority things to conduct research? In a conclusion parts, could you add a outreach of your findings for future study?

### Detail comments ###

Usually, many papers follow the order of introduction, materials and methods, results, discussion and conclusions, but do you have any intention to use a typical way?

Author Response

Dear Editors and Reviewers,

We sincerely thank for the constructive comments concerning our manuscript (plants-3540799). These comments are valuable and very helpful for improving our manuscript. Based on these comments and suggestions, we have made careful modifications to the original manuscript. The reviewer's comments are laid out below in black and have been numbered. Our point-by-point responses are given in blue and changes/additions to the manuscript are highlighted using the "Track Changes" function in the Word documents.

Comment 1

L 301~305, What is a precision and measuring range of device? Please clarify the device description.

Response 1

Thank you for highlighting the need for precise instrumentation details. We have supplemented the device specifications in Section 4.4, with key additions summarized below:

Table 1 Measurement Parameters of the sound pressure level meter

Device

Range

Measurement

Protocol

Measurement

Uncertainty

Location (Track/Clean)

AWA6228+

Sound Pressure Level Meter

f: 10Hz-20kHz

LP: 20dB-145dB

GB/T3785.1-2010

IEC61672-1:2013

0.4dB-1.0dB

Page4, Lines 159-161âž” Page 4, Lines 159-161

Question 2

Based on your data, coriaceous leaf could be an effect at the noise reduction. But, in the field application, macro-scale factors such as leaf arrangement, vacant space of twig and branches could affect the pathway of sound waves. The traits of micro-scale factors of leaf surface roughness and texture could provide a fundamental information to decrease a noise in a city. In the application of your results to a city, what is the priority things to conduct research? In a conclusion parts, could you add a outreach of your findings for future study?

Response 2

Thank you for highlighting the importance of bridging micro-scale traits and macro-scale design in urban noise mitigation. As suggested, we have expanded the Conclusion section to explicitly address this interplay (Pages 14-15, Lines 504–514 of track changes version of the manuscript)

(Page 14-15, Lines 476-483 of clean version).

Key additions include:

1.Trait-based selection:

Leaf surface roughness (LSR) and coriaceous texture (LT) are now emphasized as critical morphological filters for identifying noise-attenuating species.

2.Multi-scale integration:

We explicitly acknowledge that canopy stratification and branch porosity modulate sound scattering efficiency, requiring synergistic design beyond species selection alone.

3.Future Research Directions

3.1 Prioritize species with higher LSR and LT in high-noise zones

3.2 Coupling trait-based selection with multi-layered designs

We think that future studies should investigate the structural synergies between forest architecture, canopy morphology, and leaf trait diversity (e.g., coriaceous vs. chartaceous leaves), extending the individual-scale findings of this work to community configurations. Specifically, quantifying how micro-scale leaf adaptations (e.g., trichome density, cuticle thickness) scale up to influence macro-scale acoustic performance in heterogeneous canopies could optimize noise-attenuating forest design.

Comment 3

Detail comments

Usually, many papers follow the order of introduction, materials and methods, results, discussion and conclusions, but do you have any intention to use a typical way?

Response 3

We sincerely appreciate the reviewer’s thoughtful comment regarding the structure of our manuscript. We fully recognize the value of the traditional IMRaD format (Introduction, Methods, Results, and Discussion) in organizing scientific narratives. The manuscript now strictly adheres to the IMRaD structure prescribed in the journal guidelines.

Reviewer 2 Report

Comments and Suggestions for Authors

The subject of the paper is interesting, the research shown in the manuscript sheds light on the relationship between leaf morphological traits and noise reduction. The paper is of good quality. The methodology is appropriate. The results are presented in adequate figures and tables, and analysed using appropriate statistical methods. The Discussion is concise and highlights the key finding. The manuscript is appropriate to be published in 'Plants' but there are numerous minor points which should be improved. I have listed them in a separate file.

Comments on the Quality of English Language

Please review the text carefully, I have shown some week points in the attached file.

Author Response

Dear Editors and Reviewers,

We sincerely thank for the constructive comments concerning our manuscript (plants-3540799). These comments are valuable and very helpful for improving our manuscript. Based on these comments and suggestions, we have made careful modifications to the original manuscript. The reviewer's comments are laid out below in black and have been numbered. Our point-by-point responses are given in blue and changes/additions to the manuscript are highlighted using the "Track Changes" function in the Word documents.

Comment 1

Line 39, Keywords: “leaf texture” and “leaf surface roughness” are used in the title, so there

is no need to repeat them in the Keywords.

Response 1

Thank you for pointing this out. Following your suggestion, we have removed the keywords “leaf texture” and “leaf surface roughness” to avoid redundancy with the title.

This change has been made on Page 1, Lines 39–40 of the track-changes version of the manuscript.
In the clean version, the revision can be found on Page 1 Line39.

Comment 2

Line 96, Introduction: irrespective of the list of abbreviations added at the end of the main

text, I suggest to explain all abbreviations, when used for the first time, here: “LT”.

Response 2

Thank you for highlighting this issue. According to your suggestion, we have carefully revised the manuscript to define all abbreviations upon their first use throughout the text. These changes are explicitly marked in both versions of the revised manuscript: For a comprehensive overview, all abbreviation expansions and their locations are summarized in Table 1 (attached below).  

Table 1 Summary of Revisions: Descriptions and Version-Specific Locations for Response 2/3

Abbreviation

Full Definition

Track-Changes Version Location

Clean Version Location

LT

leaf texture

Page 3, Line 96

Page 3, Line 96

LSR

leaf surface roughness

Page 3, Line 104

Page 3, Line 104

RNR

relative noise reduction

Page 8, Line 261

Page 8, Line 262

CL

coriaceous leaf

Page 8, Line 280

Page 8, Line 281

CHL

chartaceous leaf

Page 8, Line 281

Page 8, Line 282

LSRD

leaf surface with roughness on double epidermis

Page 8, Line 284

Page 8, Line 285

LSSD

leaf surface with smooth on double epidermis

Page 8, Line 285

Page 8, Line 286

LSRS

leaves of leaf surface with roughness on single epidermis

Page 8, Line 286

Page 8, Line 287

LTD

leaf tissue density

Page 7, Line 247

Page 7, Line 249

LMA

leaf dry mass per area

Page 7, Line 239

Page 7, Line 241

SRBE

surface roughness on

abaxial epidermis of lamina

Page 6, Line 212

Page 6, Line 212

SRDE

surface roughness on

adaxial epidermis of lamina

Page 6, Line 212

Page 6, Line 211

CLSSD

coriaceous leaf with smooth

on double epidermis

Figure 8 caption, Page 6, Line 372

Figure 8 caption, Page 11, Line 369

CLSRS

coriaceous leaf with roughness

on single epidermis

Figure 8 caption, Page 6, Line 373

Figure 8 caption, Page 11, Line 370

CLSRD

coriaceous leaf with roughness

on double epidermis

Figure 8 caption, Page 6, Line 374

Figure 8 caption, Page 11, Line 371

CHLSSD

chartaceous leaf with smooth

on double epidermis

Figure 8 caption, Page 6, Line 374

Figure 8 caption, Page 11, Line 371

CHLSRS

chartaceous leaf with roughness

on single epidermis

Figure 8 caption, Page 6, Line 375

Figure 8 caption, Page 11, Line 372

CHLSRD

chartaceous leaf with roughness

on double epidermis

Figure 8 caption, Page 6, Line 376

Figure 8 caption, Page 11, Line 373

SA

sound level meter A

Page 4, Line 143

Page 4, Line 143

SB

sound level meter B

Page 4, Line 144

Page 4, Line 144

PA

area of lamina surface protrusion

Page 7, Line 217

Page 7, Line 219

HA

area of hair patches

Page 7, Line 217

Page 7, Line 219

VA

area of leaf vein

Page 7, Line 218

Page 7, Line 220

LA

leaf area

Page 7, Line 218

Page 7, Line 220

BE

abaxial epidermis

Page 7, Line 219

Page 7, Line 221

DE

adaxial epidermis

Page 7, Line 220

Page 7, Line 223

SHA

the area of the leaf surface where the hairs are located

Page 7, Line 229

Page 7, Line 231

PABE

Protrusion area on abaxial epidermis

Page 7, Line 219

Page 7, Line 221

HABE

Hair area on abaxial epidermis

Page 7, Line 219

Page 7, Line 221

VABE

Vein area on abaxial epidermis

Page 7, Line 220

Page 7, Line 222

PADE

Protrusion area on adaxial epidermis

Page 7, Line 221

Page 7, Line 223

HADE

Hair area on adaxial epidermis

Page 7, Line 221

Page 7, Line 223

VADE

Vein area on adaxial epidermis

Page 7, Line 222

Page 7, Line 224

LHP

leaf hair proportion

Page 7, Line 231

Page 7, Line 233

SPLA

sound pressure level A

Page 7, Line 252

Page 7, Line 253

SPLB

sound pressure level B

Page 7, Line 253

Page 7, Line 254

SPLA0

sound pressure level A0

Page 8, Line 257

Page 8, Line 258

SPLB0

sound pressure level B0

Page 8, Line 258

Page 8, Line 259

Comment 3

Lines 100, 104 and the next ones, Results:  please explain the abbreviations: LSR, RNR,

LSRD etc. when used in the main text for the first time

Response 3

Thank you for your valuable feedback. We have systematically revised the manuscript to ensure all abbreviations are explicitly defined upon their first occurrence in the main text. The Abbreviations and their full definition have been listed in Table 1.

Comment 4

Figure 1, 2, 3, 5: units should be inserted in parentheses, i.e. Y-axis: RNR (dB).

Response 4

Thank you for your feedback. We have inserted units in parentheses for all Y-axis labels in Figures 1, 2, 3, and 5 as requested. The modifications are detailed below with cross-referenced locations.

Table 2 Summary of Revisions of axis labels

Original

Figure

Number

New

Figure

Number

Revision Description

Track-Changes Version Location

Clean Version Location

1

3

Y-axis label updated to "RNR (dB)"

Page 9,

Lines 309

Page 9,

Line 310

2

6

Y-axis label updated to "RNR (dB)"

Page 11,

Lines 349

Page 10,

Line 348

3

7

Y-axis label revised to " RNR (dB)"

Page 11,

Lines 363-369

Page 11,

Lines 361-362

5

5

Y-axis label revised to " RNR (dB)"

Page 10,

Lines 322-323

Page 9,

Line 322-323

6

8

Y-axis label revised to " Leaf Surface roughness (%)"

X-axis label “RLS” revised to “LSR”

Page 12,

Lines 370-371

Page 11,

Lines 368

Comment 5

Lines 112-115, Figure 1 caption: where is RRNR in the graph? Please explain RLS

abbreviation.

Response 5

We appreciate your careful review. The following corrections have been implemented:

Table 3 Summary of Revisions: Descriptions and Version-Specific Locations for Response 5

Revision Description

Track-Changes

Version Location

Clean

Version Location

1

Removal of Unreferenced Term "RRNR"

Page 9, Line 311

(Figure 3 caption)

Page 9, Line 311

(Figure 3 caption)

2

Correction of "RLS" to "LSR"

Page 12, Line 371

(Figure 8 caption)

Page 11, Lines 368

(Figure 8 caption)

3

Definition Added

Leaf Surface Roughness (LSR)

Page 6, Line 377

(Figure 8 caption)

Page 11, Line 374

(Figure 8 caption)

Comment 6

Lines 125-126, Figure 2 caption: please explain all abbreviations (CL, CHL, etc.) in the

graphs.  m

Response 6

We would like to thank you for the positive and valuable comments. According to your suggestion, we have carefully modified the relevant content. The specific replies to the comments are as follows:

Table 4 Explainations on all abbreviations

Original

Figure

Number

New

Figure

Number

Revision Description

Track-Changes Version locations

Clean

Version

Location

2

6

Define all Abbreviations in the caption

Page 11

Lines

349-353

Page 10,

Lines

348-352

3

7

Define all Abbreviations in the caption

Page 11,

Lines

366-370

Page 11,

Lines

362-367

4

4

Define all Abbreviations in the caption

Page 9,

Lines

317-321

Page 9,

Lines

318-322

5

5

Define all Abbreviations in the caption

Page 10,

Lines323-331

Page 10,

Lines

324-331

6

8

Define all Abbreviations in the caption

Page 12,

Lines 371- 378

Page 11,

Lines

368-375

Comment 7

Figure 4: please explain or correct Y-axis caption: Ln-Leaf tissue density. What is “Ln”?

Response 7

We sincerely appreciate your meticulous review. We have thoroughly revised the figure captions and axis labels to ensure clarity and accuracy. The specific modifications are as follows:

  1. Figure 4: Correction of "Ln-Leaf tissue density"

Y-axis label updated: "Leaf tissue density(g·cm-3) ",the caption was amended Caption has been modified and an explanation has been added, stating that the y-axis data is the natural logarithm of leaf tissue density(Page 9, Lines 317-321 of the track-changes version of the manuscript)( Page 9, Lines 318-322 of the clean version of the manuscript).

  1. Figure 6: Correction of "Ln- Leaf surface roughness "

Y-axis label updated: " Leaf surface roughness(%) ",the caption was amended Caption has been modified and an explanation has been added, stating that the y-axis data is the natural logarithm of leaf surface roughness (Page 12, Lines 371- 378 of the track-changes version of the manuscript).(Page 11, Lines 368-375 of the clean version of the manuscript).

Comment 8

Figures 4, 5, 6 and Table 2 should be described and included in the Results. Then, in

Discussion they can be commented.

Response 8

We think this is a very insightful comment that will help advance future research and improve the quality of the paper. We have restructured the manuscript to systematically integrate Figures 4–6 and Table 2 into the Results section and expanded their discussion as recommended. The key modifications are outlined below:

Table 5 Summary of Revisions: Descriptions and Version-Specific Locations for response8

Original

Figure

Number

New

Figure

Number

Description

in

Results

Track-Changes Version Location

Clean

Version

Location

Figure4

Figure4

Meanwhile, the average LTD values of CL leaves were significantly higher than those of CHL leaves, with increases of 9.13%, 23.96%, and 25.04% under the LSSD, LSRS, and LSRD surface roughness conditions, respectively (Figure 4).

Page8,

Lines

291-293

Page8,

Lines

293-295

Figure5

Figure5

Statistical analysis of the scatter plot demonstrated a significant positive correlation between leaf tissue density (LTD) and RNR (p < 0.05), suggesting that denser leaf tissues may contribute to improved acoustic attenuation performance (Figure 5).

Page8,

Lines

287-289

Page8,

Lines

289-291

Figure6

Figure8

Notably, for both CL and CHL leaves, LSR maintained a consistent quantitative relationship across different treatment levels, with the pattern LSSD < LSRS < LSRD (Figure 8).

Page8,

Lines

299-301

Page8,

Lines

301-303

Table2

Table3

The results of Pearson correlation analysis further indicated a highly significant positive correlation between LTD and RNR (Table 3).

Page8,

Lines

289-291

Page8,

Lines

291-293

Table2

Table3

Moreover, the correlation coefficient and statistical significance between RNR and CL were notably higher than those between RNR and CHL (Table 3).

Page8,

Lines

294-295

Page8,

Lines

296-298

Table2

Table3

Pearson correlation analysis further supported this pattern, showing that the correlation coefficient between LSR and RNR was highest in LSRD, followed by LSRS, and lowest in LSSD (Table 3).

Page8,

Lines

303-305

Page8,

Lines

305-308

Comment 9

Lines 187-188, Discussion: please explain the abbreviations: SRBE, SRDE (when used for

the first time) and include them in the list of abbreviations. These parameters should be also

explained in M&M – how they were obtained/calculated.

Response 9

We are very grateful for your valuable suggestions and We have addressed the definitions, abbreviations list updates, and methodological explanations as follows:

Table 6 Summary of Revisions: Descriptions and Version-Specific Locations

Abbreviation

 Full Definition

Track-Changes Version Location

Clean Version Location

SRBE

surface roughness on

abaxial epidermis of lamina

Page 6, Line 211-212

Page 6, Line 211

SRDE

surface roughness on

adaxial epidermis of lamina

Page 6, Line 212

Page 6, Line 212

Table 7 Summary of Revisions: Descriptions and Version-Specific Locations

Version

Location

Highlight

Track-changes

Page 17, Abbreviation List

Highlighted with Yellow "SRBE/SRDE"

Clean

Page 14, Abbreviation List

Abbreviations List

 Parameter Calculation in Methods

SRBE(%)=(PABE+HABE+VABE)/LA×100%(2)                                                                                                                           

SRDE(%)=(PADE+HADE+VADE)/LA×100%(3)

Location:

Track-changes version: Page 6-7, Lines 215-218. (highlighted in yellow).

Clean version: Page 7, Lines215-218

Comment 10

Table 2: which column (line?) shows RNR?

Response 10

Thank you for your reminder! We think this is a very professional and important comment which made us realize that the original manuscript's description of Table 2 was unclear and could potentially mislead readers. Following your suggestion, we have added the following clarification to the caption of Table 3: " The data in each cell of the table represented the Pearson correlation coefficient between the structural indicators of and the RNR”

These changes can be found on Page 10,Lines 338-339 of the Track-changes version

These changes can be found on Page 10,Lines 338-339 of the clean version.

Comment 11

Line 206-206, Discussion: “because the coefficient…” this sentence is unclear.

Response 11

Thank you for highlighting the ambiguity. We have revised the manuscript to meet the request.

Original Sentence (Line 206):

"…because the coefficient between RNR and LSR on abaxial epidermis for all of data was higher than the coefficient between RNR and LSR on adaxial epidermis (Table 2). "

Revised Sentence:

"In general, leaves with LSR on the abaxial epidermis exhibited stronger noise reduction than those with LSR on the adaxial side, as indicated by the higher correlation coefficient between RNR and LSR on the abaxial surface in the complete dataset (Table 3)."

This change can be found on Page14, Lines 443-446 of the track-changes version of the manuscript.  

In the clean version, this change can be found on Page 12, Lines 428-430

Comment 12

What column in Table 2 shows RNR and LSR on abaxial epidermis or LSR on adaxial epidermis?

Response 12

Thank you for your reminder, which made us realize that the original manuscript's description of Table 2 was unclear and could potentially mislead readers. Following your suggestion, we have added the following clarification to the caption of Table 3: " The data in each cell of the table represented the Pearson correlation coefficient between the structural indicators of and the RNR.”

We have standardized the terminology usage. The data in the line labeled '(Surface roughness on abaxial epidermis of lamina) SRBE' represents the correlation coefficient between SRBE and RNR for different types of leaves. The data in the line labeled 'Surface roughness on adaxial epidermis of lamina (SRDE)' represents the correlation coefficient between SRDE and RNR for different types of leaves.

These changes can be found on Page 10,Lines 338-339 of the Track-changes version

These changes can be found on Page 10,Lines 338-339 of the clean version.

Comment 13

Main text: English language requires precise review: lines 59 (“numerical”?), 117, 128, 183,

254, 276, 410; a coma is necessary in the line 157 between “LT” and “that LTD”.

Response 13

We would like to thank you for the positive and valuable comments. We think this is a very professional and important comment. The specific replies to the comments are as follows:

Table 8 Summary of Response 13 on Revisions: Descriptions and Version-Specific Locations

Original

Line

Number

Issue

Revised

Text

Track-Changes Version Location

Clean

Version Location

59

Numerical papers of the effects of forest on noise reduction have been published.

Numerical studies on effects of forests on noise reduction have been published

Page 1,

Line 59

Page1, Line 59

117

When we compared RNR with different frequencies between CL an CHL, we can pound that RNR were remarkably reduced by CL comparing with CHL when the wave band was lower than 12 kHz (Figure 2A)

When comparing RNR across different frequencies between CL and CHL, we found that RNR was remarkably reduced by CL comparing with CHL when the waveband was below 12 kHz (Figure 6A). Furthermore, the results remained consistent with LSRD within the same frequency range (Figure 6B).

Page10, Lines

340-344

Page10, Lines

360-362

128

LSR plaid a great role in the noise reduction function of LT (Figure 3) though the interaction between LT and LSR was not much higher (Table 1).

LSR played a significant role in LT-mediated noise attenuation (Figure 7), though the level of interaction between them was relatively limited (Table 2).

Page11, Lines

355-356

Page11, Lines

354-355

157

LTD was significantly influenced by LT that LTD of CL was significantly higher than that of CHL regardless of the type of LSR (Figure 4).

LTD was significantly influenced by LT, that LTD of CL was significantly higher than that of CHL regardless of the type of LSR (Figure 4).

Page12, Lines

393-395

Page 12, Lines

390-392

183

It was reported in recent studies that thickness and density of natural fiber was the important factors to sound absorption [47], thus dense bristle on leaf surface should be efficient to noise reduction.

Recent studies have reported that thickness and density of natural fiber are the important factors in sound absorption, thus dense bristle on leaf surface should be efficient to noise reduction.

Page13, Lines

420-423

Page 12, Lines

410-412

254

The locations of sampled trees were far from large squire, main roads and water body.

The locations of sampled trees were far from large squares, main roads and water bodies.

Page3, Lines

109-110

Page3, Lines

109-110

276

Therefor the indoor experiments were conducted using a self-designed device to avoid the interference from outdoor environmental noise.

Therefore, a custom-built device was employed in indoor experiments to avoid interference from outdoor noise.

Page4, Lines

131-132

Page4, Lines

131-132

410

Noise reduction ability of woody plant is significantly affected by leaf texture, leaf surface roughness and their interaction.

Noise attenuation performance of woody plant leaves is significantly influenced by leaf texture, leaf surface roughness and the interaction between these two factors.

Page15, Lines

485-487

Page13, Lines

460-470

Comment 14

Lines 419-420: unnecessary.

Response 14

Thank you for highlighting the ambiguity. This removal was explicitly shown as a strikethrough deletion on Page 15, Lines 504-505 in the track-changes version. In the clean version, these lines have been omitted entirely (Page 14, Lines 483-484).

Comment 15

Abbreviations: why “LSR” in italics? Several abbreviations are lacking: SPLA, SPLA0, SPLB,

SPLB0, SBA, SRBE, SRDE, RLS.

Response 15

We think this is a very professional and important comment. The italicized abbreviation "LSR" in the list of abbreviations has been corrected to standard (non-italic) formatting.     The italic formatting of LSR in the abbreviation list has been corrected to standard non-italic style. This entry is located on Page 16 of the track-changes version of the manuscript,highlighted with yellow.   

Table 9 Summary of Lacking abbreviations for Response 15

Lacking

abbreviations

  Full

Definition

Track-Changes

 Version Location

Clean

Version Location

SPLA

sound pressure level A

Page 16-17

Page 14-15

SPLA0

sound pressure level A0

Page 16-17

Page 14-15

SPLB

sound pressure level B

Page 16-17

Page 14-15

SPLB0

sound pressure level B0

Page 16-17

Page 14-15

SHA

the area of the leaf surface where the hairs are located

Page 16-17

Page 14-15

SRBE

surface roughness on abaxial epidermis

Page 16-17

Page 14-15

SRDE

surface roughness on abaxial epidermis

Page 16-17

Page 14-15

CLSSD

coriaceous leaf with smooth on double epidermis

Page 16-17

Page 14-15

CLSRS

coriaceous leaf with roughness on single epidermis

Page 16-17

Page 14-15

CLSRD

coriaceous leaf with roughness on double epidermis

Page 16-17

Page 14-15

CHLSSD

chartaceous leaf with smooth on double epidermis

Page 16-17

Page 14-15

CHLSRS

chartaceous leaf with roughness on single epidermis

Page 16-17

Page 14-15

CHLSRD

chartaceous leaf with roughness on double epidermis

Page 16-17

Page 14-15

We sincerely appreciate your meticulous review. The following abbreviation inconsistencies have been thoroughly corrected in both the main text and the abbreviations list:

Table 10 Summary of correction of abbreviations

Original Incorrect Term

Revised Term

Full Definition

Track-Changes

 Location

Clean

Version Location

SBA

SHA

the area of the leaf surface where the hairs are located

Page 7, Line 229

Page 7, Line 231

HV

PA

Protrusion area

Page7, Line 217

Page7, Line 219

BA

HA

Hair area

Page7, Line 217

Page7, Line 219

RLS

LSR

Leaf surface roughness

Page 12, Line 377

(Figure 8 caption)

Page 11, Lines 368

(Figure 8 caption)
